# "I Feel Good, I Am a Part of the Community": Social Responsibility Values and Prosocial Behaviors during Adolescence, and Their Effects on Well-Being

Maria Giuseppina Bartolo * , Anna Lisa Palermiti, Rocco Servidio and Angela Costabile

Department of Culture, Education and Society, University of Calabria, 87036 Arcavacata di Rende, Italy; annalisa.palermiti@unical.it (A.L.P.); rocco.servidio@unical.it (R.S.); angela.costabile@unical.it (A.C.)
* Correspondence: mariagiuseppina.bartolo@unical.it

**Abstract:** Developing social responsibility values and a sense of community during adolescence is an important step that leads to prosocial behaviors toward others and feeling good about oneself and fellow community members. In line with the United Nations Sustainable Development Goals, sustainability is considered in a broader perspective as aimed at the development of human society, improving individual well-being and the quality of life for individuals and communities. In this sense, from a psychology of sustainability perspective, it is important to better understand the meaning of the connections between individual development and communities and the complexity of their relationships. The present study investigates the relationship between social responsibility values and well-being during adolescence, in a sample of 1925 students aged 14 to 20 years, also examining the mediating role of the sense of community and prosocial behaviors. Our mediational analysis suggests that the relation between social responsibility values and well-being is fully mediated by the sense of community and prosocial behaviors. Moreover, the sense of community has a direct effect on prosocial behaviors. This study, thus, provides new evidence and enlarges the wider sustainability science of how values and social participation enhance well-being.

**Keywords:** social responsibility values; sense of community; prosocial behaviors; well-being; sustainability; adolescence

## 1. Introduction

Adolescence is a life stage during which youth experiment with themselves in various daily circumstances by pursuing experiences beyond parental control, with peers or adults. These social and personal experiences help adolescents to face all the various developmental tasks required from society [1,2] and to engage in a wider variety of behaviors, including prosocial ones [3,4]. At the same time, these experiences improve adolescents' well-being [5,6].

Owing to the specific changes (e.g., physical, social, and psychological) that occur during adolescence, it is important to investigate how variables linked to identity development, such as social responsibility values, affect positive well-being. Social responsibility values entail taking care of and helping others; more generally, they contribute to community well-being. This is a sort of personal orientation that leads one to apply prosocial behaviors, which in turn can lead to a sense of well-being.

While it is known that values and a sense of belonging already emerge during childhood [7], in line with the Positive Youth Development (PYD) perspective, that claim that adolescents' involvement in social activism, or in initiatives to promote social changes, is the expression of an adequate growth, we suppose that the opportunities offered during adolescence contribute to making youths feel important as an active part of a community and agents of change. These opportunities help adolescents develop a sense of community that in turn encourages them to adopt prosocial behaviors which can benefit everyone, including themselves; and this has a positive impact on overall well-being.

Not least on the basis of their experiences with peers and adults (e.g., teachers, and neighbors), adolescents can develop their identity and achieve positive developmental outcomes such as well-being [8,9] through social participation [10] and civic engagement [11,12]. All of these aspects can be analyzed in the context of the broader field of sustainable development, as they highlight the importance of the relationship between people and the environment on a global scale. Sustainable development is commonly portrayed as the intersection of environmental, economic, and social issues, and its definition started from a focus on the fulfillment of basic physical needs and gradually moved to include psychological and emotional goals, such as well-being.

People create meaning and values, experience feelings of belonging to a community and adopt prosocial behaviors aimed at taking care of or helping others throughout the course of their lives. During adolescence, however, opportunities for prosocial behaviors increase [13], as there are greater chances of socialization, and more sociocognitive and socioemotive influences [14]. Therefore, it is important to investigate social responsibility values and prosocial behaviors during adolescence, particularly because they may be considered a protective factor against negative outcomes [15,16]. Moreover, starting in childhood, people develop a sense of community that becomes stronger during adolescence as youth interact with people outside their own family on a daily basis. In many cases they are also involved in volunteering and community services that benefit of other members of the community, or even strangers [17], with whom they share goals and from whom they receive positive feedback related to their helping behaviors and benefits in terms of well-being.

Decisional and behavioral processes are internal psychological mechanisms within the individual and require that they be studied and understood better from different points of view. In line with the PYD perspective, prosocial behaviors toward strangers can be regarded as a way in which adolescents contribute to their own community [18]. Such behaviors involve personal dispositions, values or willingness to help others, as well as ecological factors (e.g., the social context and emotional or physical access to resources) [19]. From a wider perspective, decisional and behavioral processes are also the result of the complexity of relationships [20,21] that are established between individuals and communities and contribute to the community's well-being in terms of sustainability, also in line with the United Nations Sustainable Development Goals (UN-SDGs) [22]. The UN-SDGs call for action by all countries and emphasize the importance of addressing the issues of equity and inequality in order to eradicate poverty, promote health, contribute to economic development, promote employment, ensure social protection, and address climate change. Southern Italy is somewhat behind in achieving these goals [23], so it is important to carry out further research in this part of the country, with a particular focus on adolescents, in order to face the challenges of sustainability.

Although there is a general consensus on the importance of adolescents' social participation, based on both research and the United Nations and World Health Organization guidelines that recognize adolescents' leading role in achieving SDGs, the Global Action for Measurement of Adolescent Health (GAMA) Advisory Group notes that it is important to use different types of indicators to measure and monitor adolescent health. The Group has highlighted the need to adopt other indicators to better evaluate—and consequently provide information about—the kind of actions that can benefit adolescents' lives [24,25].

On basis of these premises, the present study aims to test a research model that considers the effect of social responsibility values on well-being also through the mediating role of the sense of community and prosocial behaviors. We expect that the final outcome of this model is well-being understood as overall positive mental well-being that includes the affective and cognitive component of good psychological functioning.

### 1.1. Social Responsibility Values during Adolescence

Values are aspects of the self that concern personal priorities and thoughts that contribute to building one's identity and lead toward beliefs, attitudes and behaviors [26]. Social responsibility consists of a set of values or personal commitments' that involve

helping or taking care of others, including strangers, and contributing to society for the purpose of improving one's community and society. These values are generally expected to motivate a person's behaviors. With regard to social responsibility values as an orientation that motivates individuals' prosocial, moral and civic behaviors, it is possible to affirm that social responsibility values predict a broad range of prosocial behaviors, ranging from volunteering [27] to political activism [28] and pro-environmental behaviors [29].

From a psychological perspective, social responsibility is a central developmental factor during adolescence [30] and a motivator of prosocial actions [31]. Although some studies underline that values are developed during adolescence [32], and although it is widely known that during adolescence broad changes occur in the sphere of social responsibility [33–36], little attention has been paid to the role of social responsibility values during adolescence, particularly the effects of such values on well-being.

In light of this, is important to further investigate the importance of social responsibility values during adolescence as factors that foster prosocial behaviors while at the same time increase well-being.

### 1.2. The Sense of Community during Adolescence

During adolescence, youth develop their own identity and also increase relationships with peers who represent the reference group out of the family context, but at the same time, however, adolescents interact with people apart from their peers, people who are members of their community, such as neighbors, teachers and adults. Through these relationships, youth experience different social roles, adapting to the context or doing something to change it, so as to achieve personal and social well-being [2]. Youths can feel part of their community by sharing common goals and establish significant bonds with others, sharing relations, symbols or images. The feeling that emerges from the interaction between individuals and their environment [37], by taking care of the needs of one's community members and working to achieve shared goals, represent what is defined as the sense of community.

As the sense of community is based on common goals, it is reasonable to affirm that a developed sense of community translates into actions aimed at bettering common conditions. At the same time, it could mediate the relation between social responsibility values and well-being. People's involvement and commitment to solving common problems leads them to strengthen their sense community, participate in community organization, and increase their confidence in the future. Consequently, the results of this shared commitment also help to fulfil individual needs, leading to personal and community well-being. We must always consider the relationship between community well-being and individual well-being as two sides of the same coin, since they are interrelated conditions: a community will not thrive if its members are not well, and individuals cannot thrive if their community is not well [38].

Indeed, considerable evidence supports the idea that the sense of community plays a role in enhancing well-being [9,39–42]. Feeling part of a community, sharing certain values and common goals, and working to achieve personal and shared objectives increase adolescents' life satisfaction [43]. Consequently, in line with the GAMA's recommendations, well-being can be interpreted not only as the absence of illness, but as a more complex construct that is reflected in positive mental health [44–46] and includes a range of psychological, affective and social aspects [47]. The acquisition of well-being during adolescence is an important protective factor for mental health and enhances positive social adaptation [48].

### 1.3. Prosocial Behaviors during Adolescence

In line with what has been argued so far, namely that social responsibility values motivate a person's behaviors, and that the sense of community is based on common goals, values and the sense of community can be seen to translate into actions, more specifically into prosocial behaviors that are voluntary actions benefiting others, such as helpfulness, sharing, cooperativeness and empathy [49,50]. In most cases, these kinds of social behaviors are initially learned at home and applied within relationships with one's family and friends; later, they are also applied in interactions with strangers. Growing autonomy can foster

social responsibility during adolescence, and opportunities for prosocial behaviors increase in extra-familial contexts [13]. As youths build their identity and develop meanings [51], many of these behaviors are strengthened, becoming the basis for the individual's morals, identity and manner of life [52]. Moreover, research suggests that meaning is positively correlated with prosocial behaviors toward others [53,54]. These, in turn, appear to be correlated with adolescents' self-esteem [55], peer relationships [56] and social skills [57], which promote the help-giver's well-being [58,59].

Considering that prosocial behaviors are not based on close relationships [60], the motivation that drives people to implement prosocial behaviors is probably to be found in personal and social values that include an awareness of the opportunity to help others. Therefore, being aware of the effects of helping others is an important step in adolescent development [61,62].

Moreover, many previous studies underlined the negative relationship between prosocial behaviors and internalizing symptoms such as anxiety and depression [63–65]. This suggests that prosocial behaviors can have a positive effect on adolescents' well-being, an effect that derives from helping others [66]. It is reasonable to assume that in addition to being foundational markers of adolescents' social well-being [50], prosocial behaviors promote the formation and preservation of successful interpersonal relationships [67], thereby contributing to overall well-being [68,69] and developmental sustainability.

We therefore expect, on the one hand, that adolescents' social roles—more specifically, their sense of community and social responsibility values—have a positive effect on prosocial behaviors and, on the other hand, that prosocial behaviors mediate the relation between social responsibility values and adolescents' well-being.

The main aim of this study is to examine the relationship between social responsibility values and well-being during adolescence, by also examining the mediating role of the sense of community and prosocial behaviors while taking into account the empirical link between social responsibility values and well-being.

Based on the previous literature, we also expect adolescents' social participation—more specifically, their sense of community and prosocial behaviors—to have a positive effect on well-being. Since adolescence represents a critical stage in life development, playing an active part in one's community may help adolescents, during their development, to feel good about themselves and their relations with others. Helping others or acting for a common goal is equivalent to doing the "right" thing, which will make the individual feel good [70] contributing to achieving developmental sustainability.

Moreover, considering that previous studies [71] have identified differences in prosocial behaviors related to age and gender, we analyzed the model by checking these two factors.

### 1.4. Hypotheses

Overall, the present study aims to test the following research hypotheses:

**H1.** *Social responsibility values should have a direct positive effect on well-being;*

**H2.** *The sense of community and prosocial behaviors should work as serial mediators in the relationship between social responsibility values and well-being;*

**H3.** *Social responsibility values should have a positive effect on the sense of community and on prosocial behaviors.*

## 2. Materials and Methods

### 2.1. Participants and Procedures

The research was conducted through an online survey between April and May 2021. For this study, a total of 1925 adolescents (Male = 908, 47.2%; Female = 1017, 52.8%) aged 14–20 years ($M_{age}$ = 16.3, SD = 1.46) were recruited. All participants live in the South of Italy and were recruited from State High Schools in the five different provinces of the region

of Calabria through School Directors, as well as through teachers who invited students to participate in the research. As most of the participants were minors, schools had to obtain informed consent to participation from their parents. All information about the study's nature, purpose and anonymity was disseminated by teachers and was laid out at the top of the first page of the online survey. An e-mail contact was provided at the end of the survey for any questions or doubts about the research.

Participation in the study was voluntary and anonymous, and the time spent completing the online survey was about 15–20 min. The study procedures and materials were designed and employed in accordance with the ethical standards of the Italian Psychological Association (AIP).

### 2.2. Measures

The online survey comprised a battery of Italian-validated self-report scales and a socio-demographic profile.

### 2.2.1. Socio-Demographic Profile

Participants were asked to report general information about their gender, age, type of school, parental education and employment, and current living place.

### 2.2.2. Social Responsibility Value (SRV)

The social responsibility value was assessed using the Italian version of the Social Responsibility Scale [33]. This is a self-report scale consisting of 4 items; it assesses how important it is to consider other people's needs, help those less fortunate, ensure that all people are treated fairly, and consider how their actions affect people in the future (e.g., For me it is important to consider the needs of other people). The responses are given on a 5-point Likert scale, ranging from 1 (not at all important) to 5 (extremely important). For the current sample, the value of Cronbach's alpha was 0.810.

### 2.2.3. Prosocial Behaviors

Prosocial behaviors were assessed using the subscale for prosocial behaviors found in the Italian version [72] of the Strengths and Difficulties Questionnaire (SDQ) [73]. The subscale includes 5 items (e.g., Considerate of other people's feelings; Often volunteers to help others (parents, teachers, other children)) and the responses are given on a 3-point Likert scale, ranging from 0 (not true) to 2 (certainly true). Cronbach's alpha was 0.711.

### 2.2.4. Sense of Community (SoC)

The Italian version of the SoC was validated by Chiessi, Cicognani, and Sonn [74], and includes 20 items to assess the satisfaction of needs and opportunities for involvement, support and emotional connection with peers, support and emotional connection in the community, sense of belonging, and opportunities to influence the community (e.g., People in this country support each other). Each item is rated on a 5-point Likert scale from 0 (not at all true) to 4 (completely true). For our sample, the value of Cronbach's alpha was 0.929.

### 2.2.5. Warwick–Edinburgh Mental Well-Being Scale (WEMWBS)

The Italian version of the WEMWBS [75] is a measure of mental well-being that includes 12 items (e.g., I have been feeling optimistic about the future), which are positively worded in relation to each statement. Respondents are required to describe their experience over the past two weeks using a 5-point Likert-type scale ranging from 1 (never) to 5 (always). A higher WEMWBS score indicates a higher level of mental well-being. The reliability value for the present study was $\alpha = 0.88$.

### 2.3. Statistical Analyses

Statistical analyses, including descriptive statistics, univariate normality (skewness and kurtosis), and bivariate correlations (Pearson's r), were conducted using SPSS-28. No

data were missing, since responses were required for all items. Bivariate correlations were computed among the variables of interest and control variables (age and gender—point-biserial). Additionally, the reliability of scales and subscales was assessed by calculating Cronbach's alpha ($\alpha$). A structural equation model (SEM) analysis, which can simultaneously include all the independent variables, mediators, and outcome variables, was performed to test the study's hypotheses. Specifically, in the current model, SRV served as a predictor, and SoC and SDQ acted as serial mediators, and well-being was the outcome. Therefore, based on the current literature review, we were interested in testing the serial mediation effects of the sense of community and prosocial behaviors in the relationship between SRV and WB.

Given that the scales were unidimensional, solutions based on item parcelling rather than individual items seemed more appropriate to reduce the risk of convergent problems and improve model fits [76]. The parcelling was generated by applying a balanced procedure to combine high and low inter-correlation values [77]. Therefore, we consistently used three indicators for the following constructs: SoC and PWB. Moreover, direct paths from the predictors to the outcome variables were estimated, along with correlations between the control (age and gender), mediators and outcome variables. However, only the main associations were reported in the path model for the sake of clarity. The SEM model was estimated with the maximum likelihood (MLR), with standard errors and a mean adjusted chi-square test statistic robust to non-normality. To ascertain the model fit, we used the comparative fit (CFI), Tucker–Lewis (TLI) and root-mean-square error of approximation (RMSEA) indexes. Following Kline [78], we considered values of CFI $\geq$ 0.95, TLI $\geq$ 0.95 and RMSEA $\leq$ 0.05 as indications of a good model fit. SEM and mediational analyses were conducted using Mplus 7.01.

## 3. Results

Table 1 shows the descriptive statistics, indicating that participants' scores tended to cluster around the midpoint. All variables exhibited a sufficient normal distribution, with skewness and kurtosis falling within the range of +1 to –1.

**Table 1.** Descriptive statistics.

|  | **Range** | *M* | *SD* | **Kurtosis** | **Skewness** |
|---|---|---|---|---|---|
| SRV | 1–5 | 3.88 | 0.80 | 0.99 | −0.81 |
| SoC | 0–4 | 1.96 | 0.74 | 0.16 | −0.03 |
| SDQ | 1–3 | 2.46 | 0.43 | 0.13 | −0.67 |
| WB | 1–5 | 3.36 | 0.73 | 0.66 | −0.52 |
| Age | 14–20 | 16.26 | 1.46 | −0.98 | −0.96 |

Note. SRV = Social Responsibility Value; SoC = Sense of Community; SDQ = Strengths and Difficulties Questionnaire; WB = Well-Being.

Table 2 displays the correlation values among the investigated variables. As illustrated by Table 2, all the primary variables exhibited positive and significant correlations, indicating a good effect. However, certain variables, namely age and gender, were negatively correlated.

**Table 2.** Correlations between variables.

|  | **1** | **2** | **3** | **4** | **5** | **6** |
|---|---|---|---|---|---|---|
| 1. SRV | 1 |  |  |  |  |  |
| 2. SoC | 0.20 *** | 1 |  |  |  |  |
| 3. SDQ | 0.47 *** | 0.27 *** | 1 |  |  |  |
| 4. WB | 0.17 *** | 0.51 *** | 0.25 *** | 1 |  |  |
| 5. Age | −0.03 | −0.15 *** | 0.10 | −0.07 *** | 1 |  |
| 6. Gender | 0.19 *** | −0.14 *** | 0.17 *** | −0.20 *** | −0.01 | 1 |

Note. SRV = Social Responsibility Value; SoC = Sense of Community; SDQ = Strengths and Difficulties Questionnaire; WB = Well-Being. Gender (1 = male, 2 = Female) is the point-biserial correlation. *** $p < 0.001$.

The results of the SEM analysis are depicted in Figure 1. The tested model, controlling for age and gender, fits the data well, robust $\chi^2$ (105, N = 1925) = 570.56, $p$ < 0.001, CFI = 0.96, TLI = 0.95, RMSEA = 0.05, 90% CI [0.04, 0.05], SRMR = 0.04.

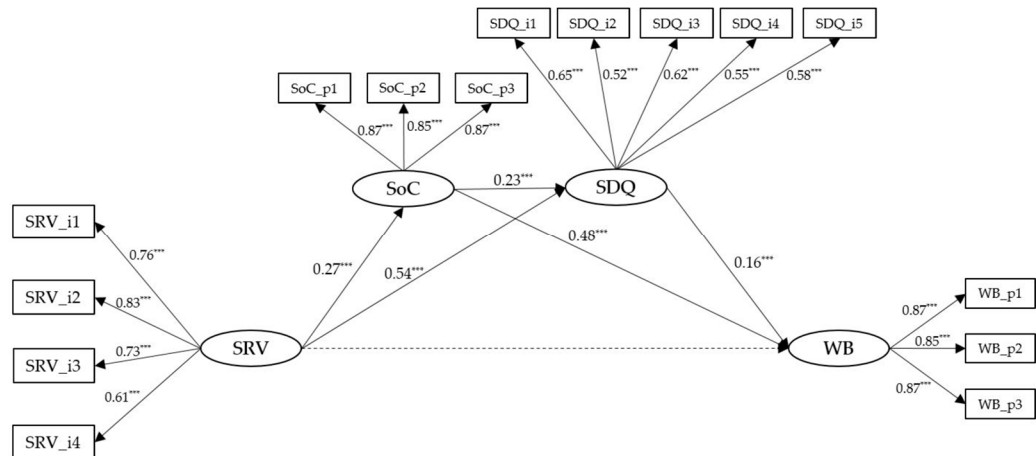

**Figure 1.** Results of the SEM model. All the values are standardized. The dashed lines indicate a non-significant path. Latent factors are presented in the circle; measured variables (parcels and single items) are presented in the rectangles. All the analyses were controlled for gender (1 = male, 2 = female) and age. Note. SRV = Social Responsibility Value; SOC = Sense of Community; SDQ = Strengths and Difficulties Questionnaire; WB = Well-Being. *** $p$ < 0.001.

The results reported in Figure 1 suggest that social responsibility values directly and positively affect the sense of community, β = 0.27, $p$ < 0.001, and strengths and difficulties, β = 0.54, $p$ < 0.001. In turn, the sense of community, β = 0.48, $p$ < 0.001, and strengths and difficulties, β = 0.16, $p$ < 0.001, positively affect well-being. Moreover, no significant direct effects emerged between social responsibility values and well-being ($p$ > 0.05).

The results of the mediational analysis indicate that the sense of community and strengths and difficulties fully mediate the association between social responsibility values and well-being. The results of the mediational analysis are reported in Table 3.

**Table 3.** Mediation and indirect effects with standardized estimates of SRV (Social Responsibility Value; independent variable), SoC (Sense of Community, SDQ, Strengths and Difficulties Questionnaire, mediators), and WB (Well-Being, outcome).

| Pathway | Estimate | SE | z | $p$ |
|---|---|---|---|---|
| SRV–>WB | | | | |
| Total effect | 0.24 | 0.03 | 7.64 | 0.000 |
| Direct effect | 0.02 | 0.04 | 0.44 | 0.660 |
| Specific indirect effects | | | | |
| SRV–SDQ–WB | 0.09 | 0.02 | 3.69 | 0.000 |
| SRV–SoC–WB | 0.13 | 0.02 | 7.90 | 0.000 |
| SRV–SDQ–SoC–WB | 0.01 | 0.00 | 3.50 | 0.000 |

## 4. Discussion

The present study explored the relationship between social responsibility values and well-being during adolescence, more specifically by identifying the mediating role of the sense of community and prosocial behaviors as factors having a positive effect on well-being.

The correlation between social responsibility values and well-being was significant, but the direct effect was no longer significant in the mediation model, so hypothesis 1 was only partially confirmed: this relation was fully mediated by a sense of community and prosocial behaviors, in accordance with hypothesis 2. Furthermore, the sense of community directly affects prosocial behaviors.

Social responsibility values and the sense of community are two closely intertwined dimensions of the social roles that adolescents can assume in society, and they directly or indirectly contribute to well-being, also through prosocial behaviors associated with social participation that help adolescents to feel well [79].

In line with WHO [80], well-being is defined as "a positive state of affairs, brought about by the simultaneous and balanced satisfaction of diverse objective and subjective needs of individuals, relationships, organizations, and communities" [81]. Therefore, it is influenced by people's values and goals, and regards the evaluation of one's own contributions to society and the achieve of the common good by working with others. In light of our results, it is possible to affirm that social responsibility values have an impact on prosocial behaviors (hypothesis 3), which in turn lead to well-being. Our findings also confirmed an indirect significant association between the two through the mediating role of the sense of community. In particular, adolescents who feel more involved in their community are engaged in prosocial behaviors and hence have higher levels of well-being.

Previous researchers have investigated prosocial behaviors with the aim of better understanding these types of behaviors [82] and their target [83–85]. In many cases, however, prosocial behaviors have not been studied by taking into consideration their origin and positive effect on well-being. This study improves our knowledge by considering prosocial behaviors as factors strongly related to social values (e.g., responsibility, and sense of community) and their positive effects on adolescents' well-being. What determines a person's level of involvement in prosocial behaviors is the result of his or her culture, situation, and motives. For example, as adolescents gain autonomy and develop their identity, they also develop social responsibility values and a sense of community that lead them to adopt prosocial behaviors. Motivations such as moral principles or values lead them to engage in behaviors such as helping strangers [84,85], taking peoples' needs into consideration, and not harming others.

Considering that some researchers affirm that prosocial behaviors are associated with positive psychological outcomes such as less externalizing behaviors [86], we supposed—and our findings confirmed—that prosocial behaviors positively affect well-being. Our study demonstrates that doing good for others benefits the givers by increasing their level of well-being. Our results can be also analyzed considering the psychology of sustainability and sustainable development as they offer their contribution to point out the well-being as outcome of relationships and behaviors in communities.

Moreover, our findings give additional support to previous work suggesting that prosocial behaviors like volunteering and civic engagement may be an important protective factor because of their influence on self-perception [87], which is an important target for many successful youth interventions [88].

Considering the current results, is seems possible to obtain positive results by applying the educational program in schools, as this is the best place to work with adolescents. Some evidence from educational projects applied in the Italian context [89–91] shows that the implementation of projects based on active citizenship, the promotion of prosocial behaviors, and active participation in the pursuit of a common goal have positive effects in terms of sociality and lower levels of aggression, contributing to a better school climate and hence to individuals' well-being.

Our study, which highlights the importance of reinforcing social responsibility values and a sense of community through educational approaches, suggests that direct social policies should not only consider adolescents a future generation targeted for the evaluation of the achievement of well-being and sustainability. Rather, adolescents should be considered as individuals in which to invest to improve sustainability goals through appropriate strategies aimed at fostering prosocial behaviors, participation in youth mutualism projects [92], and the regeneration of the political landscape [93].

## 5. Conclusions

Adolescence is a central area of concern for developmental psychology, but it also involves other fields linked to the environment, individual relationships and the role people play in their community, with a broader perspective linked with sustainable development and psychology of sustainability.

Through the present study, we intended to contribute to the current research by highlighting the effects of social participation on well-being among adolescents. By investigating a sample of Italian adolescents, we found that social participation has an impact on well-being during adolescence.

In a preventive perspective, encouraging adolescents to actively engage in social participation, do something beneficial to their community or share certain goals with their community can help them create meanings and shared values that, through prosocial behaviors, can in turn lead them to attain psychological well-being and achieve sustainable development. The results of this study can help educators, teachers, researchers, and policy makers to better target policies and interventions to face future changes, respond to challenges, and achieve sustainable development goals. Indeed, it is only through direct knowledge about a community (its needs, meanings, behaviors, beliefs, etc.) that it is possible to track change [94].

Despite the positive results obtained by this research, gaps and limitations must be taken into consideration. Firstly, although we had a relatively large sample, the study was developed during the COVID-19 pandemic, so we recruited a convenience sample only in one region of Southern Italy. Future studies should examine adolescents from other regions for generalizability to the broader population of adolescents. Moreover, in future studies, it would be interesting to take into account cultural differences not only between southern and northern Italy, in order to compare the meaning of certain results within the same State, but also with other countries, in order to verify whether cultural differences may have an impact on different types of social involvement. Secondly, the study uses self-report measures that are vulnerable to desirable responses and bias. Multiple evaluation methods would be beneficial in future researches. Thirdly, although some socio-demographic information (e.g., parental education and employment) was gathered, it was not the focus of this research. However, such information could be used in future studies. Finally, as the study focused only on general variables, future studies should pay greater attention to specific forms of prosocial behavior.

Despite these limitations, the present study contributes to the literature on adolescent development by underlining the importance of the role of social participation in young people's development.

**Author Contributions:** Conceptualization, M.G.B. and A.L.P.; methodology, M.G.B., A.L.P. and R.S.; investigation, M.G.B.; data curation, R.S.; writing—original draft preparation, M.G.B. and R.S.; writing—review and editing, M.G.B. and A.L.P.; supervision, A.C.; funding acquisition, M.G.B. All authors have read and agreed to the published version of the manuscript.

**Funding:** This research was funded by REGIONE CALABRIA, PAC Calabria 2014–2020–Asse Prioritario 12, Azione B) 10.5.12 Area Tematica S3 regionale "Turismo e Cultura".

**Institutional Review Board Statement:** All subjects gave their informed consent for inclusion before they participated in the study. The study was conducted in accordance with the Declaration of Helsinki and the ethical standards of the Italian Psychological Association (AIP).

**Informed Consent Statement:** Informed consent was obtained from the parents of all subjects involved in the study.

**Data Availability Statement:** The data presented in this study are available upon request from the corresponding author.

**Conflicts of Interest:** The authors declare no conflict of interest. The funders had no role in the design of the study; in the collection, analyses, or interpretation of data; in the writing of the manuscript; or in the decision to publish the results.

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
