# Peer review of "“I Feel Good, I Am a Part of the Community”: Social Responsibility Values and Prosocial Behaviors during Adolescence, and Their Effects on Well-Being"

_sustainability, doi:10.3390/su152316207_

Round 1
Reviewer 1 Report
Comments and Suggestions for Authors
Dear authors,
Very interesting manuscript, but I have some recommendations to improve the article. It would be useful to formulate the scientific problem in the introduction and abstract, which would later lead to clearer conclusions. Although the list of used scientific literature is very long, the used scientific literature of 2023, 2022, 2021 is missing (actually 1 per mentioned year). Although you mentioned the possible limitations of the research in the conclusions, I would think that there may be another one not mentioned - cultural differences. We have to admit that the population of Italy is much more communicative than in the northern regions, where different results would probably be obtained. What do you think?
good luck
Author Response
Dear Reviewer , thank you for your constructive feedback and suggestions. All changes in the manuscript have been made in red.
In attachment the reply to your questions.
Regards

Reviewer 2 Report
Comments and Suggestions for Authors
Overall, it is a good study on social responsibility values and how these affect positive psychological well-being.
The connection of this study with the UN-SDG and community well-being (CWB) would need more attention to the best literature in this area that inherently relates to the topics addressed in the study.
Specifically on community well-being, see studies like VanderWeele, T. J. (2019). Measures of community well-being: a template. International Journal of Community Well-Being, 2(3-4), 253-275. https://doi.org/10.1007/s42413-019-00036-8
See also Christakopoulou, S., Dawson, J. & Gari, A. The Community Well-Being Questionnaire: Theoretical Context and Initial Assessment of Its Reliability and Validity. Social Indicators Research 56, 319–349 (2001). https://doi.org/10.1023/A:1012478207457.
The connotation of the study to the UN-SDGs is too vague. Consider rewrite the introduction of these core arguments with background studies that would provide a more explicit context like:
Klein, J. D. (2021). Adolescent Health Measurement—A Necessary Step Toward Achieving Global Goals. Journal of Adolescent Health, 68(5), 836-839.
Shaw, N. (2023). Social and emotional well-being in a turbulent world. In Inclusive Education in the Early Years: Right from the Start (pp. 167-176).
Some of the landing links in the endnotes are not the correct ones. For example, see the link for Note 30 that would like this instead: https://psycnet.apa.org/fulltext/2015-53531-001.html
A few other notes:
Line 209-210: Inconsistency in the name: Warwick-Edinburgh Mental Well- 523 Being Scale (WEMWBS), which includes 14 items even in the cited Italian article.
Line 74 and note [22] in Line 415: In the English version is United Nations (UN). Please verify the link.
Several DOI Links do not work, including:
Law, B.M.; Shek, D.T. Beliefs about volunteerism, volunteering intention, volunteering behavior, and purpose in life among Chi- 479 nese adolescents in Hong Kong. The Scientific World Journal 2009, 9, 855–865. https://doi.org/ 10.1100/tsw.2009.32
20. Di Fabio, A. and Blustein, D. L. (Eds). Ebook Research Topic From Meaning of Working to Meaningful Lives: The Challenges ofExpand- 409 ing Decent Work. Frontiers Media: Lausanne, 2016. https://doi.org/10.3389/978-2-88919-970- 9 (Ebook).
Comments on the Quality of English Language
A few errors here and there. A good editing will improve the article.
Author Response
Dear Reviewer , thank you for your constructive feedback and suggestions. All changes in the manuscript have been made in red.
In attachment the reply to your questions.

Reviewer 3 Report
Comments and Suggestions for Authors
This is a good study with well designed and strong data and evidence support.It might be better if there are more discuss to build the theoretical framework. Further policy suggestions need to offer by the end of the discussion part.
Could be accepted after minor modification.
Author Response

(The authors gave the same response as above.)

Reviewer 4 Report
Comments and Suggestions for Authors
Dear author(s),
Your manuscript's topic is extremely interesting and up to date but there are a few improvement suggestions, I would kindly ask you to see below:
- line 36- I would replace we assume with it was already acknowledged
- line 67- What is PYD- maybe it would be appropriate to write the extensive name and then use the abbreviation.
- I strongly believe that the Introduction section can be improved by adding a information about the context of the manuscript, i.e. some information about Italy, and maybe the South of Italy.
- line 98- increasing should be increase
- line 124- in line with what has been argued so far, you need to add what
- again the Literature review section should be improved with some information about Italy with examples for prosocial behaviours for adolescents which worked.
- lines 164-165 need a reference or references.
- lines 185-186- the author(s) collected many socio-demographic information but used only gender and age in the analysis. What was the relevance of other socio-demographic information such as type of school, parental education and employment or current living status? Maybe the author(s) may add a few information about the respondents' demographic profile (%....were in public schools, %...were in private schools, etc) or mention them in the limitations section.
- regarding the statistical analyses subsection, I would have started with a check of the normality of the data with skewness and kurtosis.
- in line 217 I would also recommend the authors(s) to mention point biserial correlation along Person's r
- I highly recommend the author(s) to include a hypothesis subsection, for instance 2.4., where to mention the hypotheses of the model
- the discussion section should be improved. The relevance of the study for the Italian population. Since in line 280 there was an unexpected result, maybe some examples of prosocial behaviour for adolescents which improved their wellbeing would support the main outcome. Thinking a bit about discussing the author(s) results in a more applicable context, by providing examples from the Italian adolescent population.
Thank you!
Good luck!
Author Response

(The authors gave the same response as above.)

Reviewer 5 Report
Comments and Suggestions for Authors
Author Response
Dear revisor, thank you for your feedback and suggestions.
In the revised manuscript, we have made revisions in response to the concerns that were raised. All changes in the manuscript have been made in red.
